# Biomarkers and Mechanisms of Oxidative Stress—Last 20 Years of Research with an Emphasis on Kidney Damage and Renal Transplantation

**DOI:** 10.3390/ijms22158010

**Published:** 2021-07-27

**Authors:** Karol Tejchman, Katarzyna Kotfis, Jerzy Sieńko

**Affiliations:** 1Department of General and Transplantation Surgery, Pomeranian Medical University, 70-111 Szczecin, Poland; ktej78@pum.edu.pl (K.T.); jsien@poczta.onet.pl (J.S.); 2Department of Anesthesiology, Intensive Therapy and Acute Intoxications, Pomeranian Medical University, 70-111 Szczecin, Poland

**Keywords:** oxidative stress, biomarkers, antioxidants, lipid peroxidation, protein peroxidation, DNA peroxidation, signaling pathway, kidney, renal transplantation, ischemia-reperfusion injury

## Abstract

Oxidative stress is an imbalance between pro- and antioxidants that adversely influences the organism in various mechanisms and on many levels. Oxidative damage occurring concomitantly in many cellular structures may cause a deterioration of function, including apoptosis and necrosis. The damage leaves a molecular “footprint”, which can be detected by specific methodology, using certain oxidative stress biomarkers. There is an intimate relationship between oxidative stress, inflammation, and functional impairment, resulting in various diseases affecting the entire human body. In the current narrative review, we strengthen the connection between oxidative stress mechanisms and their active compounds, emphasizing kidney damage and renal transplantation. An analysis of reactive oxygen species (ROS), antioxidants, products of peroxidation, and finally signaling pathways gives a lot of promising data that potentially will modify cell responses on many levels, including gene expression. Oxidative damage, stress, and ROS are still intensively exploited research subjects. We discuss compounds mentioned earlier as biomarkers of oxidative stress and present their role documented during the last 20 years of research. The following keywords and MeSH terms were used in the search: oxidative stress, kidney, transplantation, ischemia-reperfusion injury, IRI, biomarkers, peroxidation, and treatment.

## 1. Introduction

Oxidative stress is a complex phenomenon adversely influencing the organism in various mechanisms and on many levels. It is defined as an imbalance between pro-oxidants and a network system of antioxidant defenses. In the assessment, we can measure specific biomarkers on a molecular and cellular level, observe characteristic microscopic changes in tissues, and finally diagnose certain pathologies in organs influencing the whole organism. The constant triad of oxidative stress, inflammation, and functional impairment has been reported in the pathogenesis of many diseases and clinical conditions. Oxidative stress influences aging, carcinogenesis, and metabolic syndrome, including diabetes and cardiovascular conditions [1,2].

During renal transplantation, oxidative stress is a crucial mechanism adversely affecting the kidney allograft in the ischemic phase of organ preservation and during reperfusion when sudden oxidation makes the graft prone to additional damage. This phenomenon, broadly described as ischemia-reperfusion injury (IRI), results in cell energy depletion and an imbalance in favor of pro-oxidants with microcirculatory impairment, inflammation, and apoptosis [3,4,5]. IRI was the foundation of this review. However, despite the transplantation context, we tried to present a broader picture of basics, present knowledge, and modern approaches. From the transplantation point of view, oxidative stress is a key mechanism in IRI, where both IRI and acute rejection (AR) are major causes of graft dysfunction and loss [6,7]. Adequate cellular level interventions and solutions might improve the outcomes after transplantation and help overcome the growing number of patients on transplant waiting lists. In general, studies regarding oxidative stress mechanisms and their signaling pathways give a lot of promising data. It is possible that in the future, researchers and clinicians will be able to modify cell responses on many levels, including gene expression.

In the current narrative review, we try to bring closer the oxidative stress mechanisms and their major contributors—reactive oxygen species (ROS), which also serve as biomarkers. The link between oxidative stress and IRI, inflammation, and kidney damage is presented. Moreover, we present antioxidants as well as peroxidation products and their role as biomarkers. Finally, we discuss a link between oxidative stress, signaling pathways, and potential therapeutical options. The following keywords and MeSH terms were used in the search: oxidative stress, kidney, transplantation, ischemia-reperfusion injury, IRI, biomarkers, peroxidation, and treatment.

## 2. Oxidative Stress

Oxidative stress is frequently defined as an imbalance between pro-oxidants and antioxidants [8]. It arises when the production of reactive oxygen species (ROS) overwhelms the intrinsic antioxidants. Living cells are under constant oxidative attack from ROS, leading to “oxidative damage”, and the complex antioxidant defense system generally holds this attack in balance [9]. The regulation of redox (reduction and oxidation) state is critical for cell viability, activation, proliferation, and organ function. A pathological shift in that balance leads to growing ROS concentrations, resulting in adverse modifications to cell components, such as lipids, proteins, and DNA [10]. The balance is maintained either by enzymatic antioxidants, which will be discussed further due to their role as biomarkers, or by non-enzymatic antioxidants, which compose the total antioxidant capacity (TAC) and indicate the cells’ ability to counteract oxidative stress-induced damage. TAC is greatly supported by reduced and oxidized forms of glutathione GSH/GSSG. However, the equilibrium is maintained by many other chemicals, and it seems to be regulated through discrete redox pathways rather than through direct response to chemical toxicants and physiologic stimuli. Thus, oxidative stress may also be defined as a disruption of redox signaling and control [8]. Minor disturbances lead to homeostatic adaptations, whereas significant perturbations may lead to irreparable damage and cell death [9]. Oxidative stress activates multiple intracellular signaling, which induces apoptosis or cell overgrowth, leading to organ dysfunction of the heart, pancreas, kidneys, and lungs, further causing hypertension, diabetes, chronic kidney disease, and pulmonary disorders [11]. Various pathways are involved including apoptotic genes: caspase-3, -8, -9, Bim, Bcl-2, Bak, and Bax; and oxidative stress genes: CYGB (cytoglobin), GSTP1 (glutathione S-transferase pi 1), NCF1 (neutrophil cytosolic factor 1), GPX1 (glutathione peroxidase 1), SOD1 (superoxide dismutase 1), SOD2, CCS (copper chaperone for superoxide dismutase), and NOS2 (nitric oxide synthase 2) [12]. GSTP1 expression and apoptotic signaling through activation of c-Jun N-terminal kinase (JNK) seem to be mechanisms linking oxidative stress and hypertension in spontaneously hypertensive rats [13].

Oxidative stress is recognized as a risk factor for various adverse events, including atherosclerosis and mortality in chronic kidney disease (CKD) patients. From the early stages of the disease, oxidative stress accompanies the deterioration of renal function, which is further aggravated by hemodialysis [14,15,16]. Data suggest that cardiovascular complications in patients undergoing hemodialysis are aggravated by oxidative imbalance, which may be a potential target for therapy [17]. Later, recipients enter the kidney transplantation procedure (KTx) with impaired homeostasis, further altered by preoperative and postoperative factors. Oxidative stress and oxidative damage are recognized to be important factors in development various diseases: Alzheimer’s disease [18], atherosclerosis [19], male infertility [20], COPD [21], Glaucoma [22], chronic inflammation and amyloidosis [23], Parkinson disease [24], obesity [25,26,27,28], and diabetes [29] as well as aging [30]. If oxidative damage contributes significantly to disease pathology, then actions that decrease it should be therapeutically beneficial. Some of those are discussed further.

Oxidative stress is directly caused by ROS. In normal equilibrium, they play important roles as second messengers in many intracellular signaling cascades aimed at maintaining the cell in homeostasis with its immediate environment. They cause indiscriminate damage to biological molecules at higher levels, leading to loss of function and cell death. Biomolecules in living organisms are highly exposed to oxidative stress. ROS are produced from molecular oxygen due to normal cellular metabolism; however, many other factors alter this process. Some of them are presented in Figure 1.

ROS can be divided into two groups: free radicals and non-radicals. Free radicals are molecules containing one or more unpaired electrons (^•^), which gives them high reactivity. ROS that share their unpaired electrons are non-radical forms. They have important chemical differences yet share similar mechanisms for damage at the level of biomolecules [31]. Major ROS that are of physiological significance from the first group are superoxide anion (O_2_ + e → O_2_^•−^), hydroxyl radical (H_2_O_2_ + e → OH^–^ + OH^•^), and hydroperoxyl radical (O2^•−^ + H_2_O→H_2_O^•^); from the second group—hydrogen peroxide (H_2_O^•^ + e + H → H_2_O_2_) [10]. All oxygen radicals are ROS, but not all ROS are oxygen radicals. Even though they are particles of usually short half-life, they have long been recognized as chemicals with important dual roles. They cause cellular damage by reacting with biomolecules, but they also function as cellular signaling agents [32]. Reactive species (RS) derived from molecular oxygen (ROS) and nitrogen (RNS) have been deeply studied; however, new radical species such as chlorine (RCS), bromine (RBS), and sulfur-derived species have also been identified [33].

Major ROS are presented in Table 1. Descriptive characteristics were extended by histograms representing research interest in the PubMed database. “General” interest was determined by search formula: {“ROS”[Title]}. Research interest in the field of transplantation, as a Medical Subject Headings, was determined by search formula: {(“ROS”) AND (transplantation [MeSH Major Topic])}. We can observe that peak interest has passed for ROS such as superoxide anion, alkoxyl & peroxyl, nitric oxide, and peroxynitrite; however, there are still broadly published articles about hydroxyl radical, hydrogen peroxide, singlet oxygen, ozone, hypochlorous acid, and nitrogen dioxide. There was a noticeable interest in superoxide anion, hydrogen peroxide, and nitric oxide in transplantation; however, significantly decreasing as we speak of the role of reactive oxygen species. There are more ROS known; however, they are less represented in the research.

In the group of free radicals, the following should be mentioned: carbonate (CO_3_^•−^), carbon dioxide (CO_2_^•−^), atomic chlorine (Cl^•^). In the group of non-radicals: peroxynitrous acid (ONOOH), nitryl (nitronium) chloride (NO_2_Cl), chloramines, chlorine gas (Cl_2_), nitrous acid (HNO_2_), nitrosyl cation (NO^+^), nitroxyl anion (NO^–^), dinitrogen trioxide (N_2_O_3_), dinitrogen tetraoxide (N_2_O_4_), nitryl chloride (NO_2_Cl), nitronium (nitryl) cation (NO_2_^+^), alkyl peroxynitrites (ROONO).

## 3. Oxidative Stress in Kidney Transplantation

### 3.1. Oxidative Stress in Ischemia-Reperfusion Injury

Ischemia/reperfusion injury (IRI) gained significance due to ubiquitous adverse concern in every transplantation proceeding. Damaging effects regard both reperfusion and ischemia, and they are additive. In general, IRI describes functional and structural changes that become apparent during both phases. Various molecular mechanisms have been proposed to explain IRI, however oxidative stress and ROS generation continue to receive much attention as key factors in the pathogenesis [34]. The first ultra-structural manifestation of ischemia is edema, which macroscopically is expressed by paleness and an increase in turgor and organ weight. At the molecular level, it depends on tissue hypoxia and consequent depletion of cellular ATP. Ischemic injury results in systemic inflammation due to cytokine production and increased expression of adhesion molecules by hypoxic parenchymal and endothelial cells [35]. Sudden reintroduction of O_2_ to hypoxic tissue results in an additional unique type of injury, which is not present during the ischemic phase. Forty years have passed since ROS were first pointed out in IRI. The evidence was based on three lines: (1) ROS scavengers protect against IRI, (2) artificial ROS generation resembles the IRI response, and (3) post-ischemic tissues are characterized by the enhanced ROS production and their products. The early studies were performed on SOD both in vivo and ex vivo IRI models, next CAT, GPx, and the role of H_2_O_2_ as an active metabolite, finally as a signaling second messenger [36]. The connection between ROS generation and IRI was determined based on peroxidation products as biomarkers. Most of the further discussed biomarkers were studied, with the most “popular” being MDA, 4-HNE, protein carbonyls, 3-nitrotyrosine, and 8OHdG.

Oxidative stress following IRI was also linked to the other non-enzymatic sources: hemoglobin and myoglobin. However, those enzymatic or enzyme-related processes were the most documented, including xanthine oxidase, NADPH oxidase, mitochondria, NOS, cytochrome P450, lipoxygenase/cyclooxygenase, and monoamine oxidase. Finally, oxidative stress was linked to IRI by cellular signaling and activation of certain metabolic pathways and genes. Reoxygenation mobilizes neutrophils, CD4^+^ T lymphocytes, platelets. Activated cells produce ROS, TNF-α, and inflammatory mediators [37]. Re-oxygenation increases the amount of ROS in the parenchymal, endothelial, and lymphocytic cells. Damaged mitochondria are characterized by the incomplete reduction of oxygen, resulting in superoxide anions production. There are reduced levels of NO leading to vasoconstriction, accompanied by increased expression of adhesion molecules [4]. The causes and impact of oxidative stress on the kidney, presented in Figure 1, are universally shared, including chronic kidney disease, IRI in KTx, kidney graft in the long-term outcome, other pathologies as well as healthy kidneys at normal function. The differences come from additional pathomechanisms related to the particular phase of the transplantation procedure or clinical state. Renal ischemia-reperfusion (I/R) injury is a major cause of acute kidney injury (AKI), which in KTx usually manifests as delayed graft function (DGF) [38]. The early factors that influence the late outcome of KTx take place in the donor and even long before the patient is considered a donor. It is well known that organs with extended criteria are more susceptible to ischemia-reperfusion injury (IRI). A lot of factors and their consequences can be linked to oxidative stress and chronic inflammation, with a major post-transplant determinant—delayed graft function (DGF)—as well as related complications including autoimmunological rejection (AR) [6,35,39].

### 3.2. Oxidative Stress and Inflammation

Oxidative stress can activate various transcription factors, which lead to the differential expression of some genes involved in inflammatory pathways. The main targets of oxidative stress are proteins, lipids, and DNA/RNA, which will be discussed further. Oxidative damage causes certain modifications to molecules triggering a complex response of various metabolic and signaling responses.

Inflammation is a natural defense mechanism against pathogens, and it is associated with many pathologies: infections, radiation, toxins, and diseases. There is much evidence that oxidative stress and systemic inflammation are the coexisting phenomena influencing each other. Glutathione (GSH) reduction positively correlates with the increase of oxidative stress and takes part in the redox regulation of immunity [40]. Inflammatory stimuli induce the release of a ubiquitous redox-active intracellular enzyme (PRDX2), which acts as a redox-dependent inflammatory mediator activating macrophages to produce pro-inflammatory TNF-α [32]. Chronic inflammations were reported to increase lipid peroxidation products, nitrite levels, and malondialdehyde (MDA) [41]. Oxidative stress was also reported to elevate the level of proinflammatory interleukin-6 (IL-6), vascular cell adhesion molecule-1 (VCAM-1), intercellular adhesion molecule-1 (ICAM-1), and nuclear factor-kappa B (NF-κB) [42]. The activation of the inflammatory response also leads to the activation of cellular adhesion receptors. Neutrophils migrate through the endothelial wall into the tissue parenchyma releasing cytotoxic mediators such as TNF, interleukins (ILs), and NOS, directly or indirectly leading to the production of highly reactive ROS: O_2_^•−^, H_2_O_2_, and ONOO^–^ [43]. The increase in the levels of circulating ROS and proinflammatory cytokines induces oxidative stress and inflammation in distant organs [4].

### 3.3. Oxidative Stress and Kidney Damage

The presence of ROS in biological tissues leads to a harmful oxidation effect on all their biochemical components: lipids, proteins, carbohydrates, and nucleic acids. Thus, it also plays a role in the pathophysiology of renal impairment and is a mediator of chronic kidney disease progression [44]. Oxidative stress and ROS generation in the kidney disrupt the excretory function of each section of the nephron. It impairs water–electrolyte and acid–base balance and affects kidney regulatory mechanisms: tubular glomerular feedback, myogenic reflex in the supplying arteriole, and the renin–angiotensin–aldosterone system [45]. Oxidative stress is directly linked to podocyte damage (edema, apoptosis, and necrosis), depressed glomerular filtration rate, proteinuria [46], and tubulointerstitial fibrosis [47]. Adverse metabolic changes are synergistically linked with alterations in renal hemodynamics [48]. Podocytes are vulnerable to oxidative damage. The consequence of an injury is proteinuria [49], which becomes an essential factor in inducing mesangial and tubular toxicity and is involved in local and systemic inflammatory pathways [50]. Inflammation and TGF-β are involved in podocytes’ endothelin signaling, which suppresses mitochondrial function and induces oxidative stress in the glomerular endothelium [51]. After initial kidney injury, repair mechanisms, growth factors, cytokines, and specific molecular pathways lead to tubulointerstitial fibrosis, deposition of an interstitial matrix with inflammatory cells, tubular cell loss, fibroblast accumulation, and rarefaction of the peritubular microvasculature [52]. Kidney damage is enhanced by upregulation of NOX synthesis [53], Nrf2/Keap1 system [54], and unbalance in autophagy signaling [55]. Oxidative stress is also related to endothelial dysfunction and plays a critical role in CKD progression [56]. The crucial factor is nitric oxide (NO), which is involved in several biological processes, including vasodilatation in smooth muscle cells, inflammation, and immune responses [57]. Microvascular dysfunction in oxidative stress kidney damage is mediated through nitric oxide synthase (NOS), impairment of the renal afferent arteriole autoregulation [58], increase in perfusion pressure, which increases the amount of superoxide radical (O_2_^•−^) [59].

IRI causes structural and functional damage of renal tubules by directly inducing the death of tubular cells, which further may trigger damaged responses [60]. Abnormal apoptosis and endoplasmic reticulum stress (ERS) of renal tubular epithelial cells may affect the occurrence and progression of acute kidney injury (AKI) [61].

## 4. Biomarkers of Oxidative Stress

Reactive oxygen species are compounds that are difficult to measure when assessing oxidative stress, primarily due to the very short half-life, so they hardly play the role of biomarkers. However, if ROS combines with a particular biological molecule, it leaves a unique chemical “fingerprint”. Biomarkers obtained that way can be used to evaluate oxidative damage or the effects of antioxidants, including therapeutic agents. The core criterion for the biomarker is its role in the prediction of the later development of disease. Moreover, important technical criteria of a biomarker are it should detect a major part of total ongoing oxidative damage in vivo, should provide coherent laboratory assays, results should not vary under the same conditions, should be stable during storage, must employ chemically robust measurement technology, and must not be confounded by diet [31]. There is no ideal biomarker, yet many provide sufficient accuracy. ROS, as highly reactive substances, interact with the environment in vivo, involving and stimulating various endogenous mechanisms as well as react with numerous molecules, leaving a mentioned fingerprint, which becomes the point of interest in specific evaluations. ROS, reactions, and essential antioxidants were presented in Figure 2. The nuclear signaling mechanisms are mentioned in Section 5.

### 4.1. Endogenous Antioxidants

ATP cell production is inherently connected with oxidation, reduction, and ROS generation. External factors involve microbial infections, xenobiotics, diet toxins, radiation, environmental pollution, and others. Living organisms developed specific defense systems against the deleterious action of free radicals. The most important mechanisms are intracellular; however, they act with both extracellular and dietary exogenous antioxidants. Endogenous antioxidants are divided into two groups: protein (with enzymatic activity) and non-protein. Protein ones are the first line of defense, with three the most important: CAT, SOD, and GPx. PubMed search formula for general interest was determined by: {(“biomarker”[Title/Abstract]) AND (oxidative stress)}. Research interest in the field of transplantation, as a Medical Subject Headings, was determined by search formula: {(“biomarker” [Title/Abstract]) AND (transplantation)} (Table 2).

Catalase (CAT), a tetrameric porphyrin-containing enzyme found in nearly all living organisms exposed to oxygen, is located mainly in peroxisomes. Conversion of H_2_O_2_ to water and molecular oxygen takes place in two steps: (1) CAT-Fe (III) + H_2_O_2_ → H_2_O + O=Fe(IV)-CAT(^•+^) and (2) O=Fe(IV)-CAT(^•+^) + H_2_O_2_ → CAT-Fe(III) + 2H_2_O + O_2_. The highest activity of CAT appears to be in the liver and erythrocytes [62]. It can also catalyze the oxidation, by hydrogen peroxide, of various metabolites and toxins, including formaldehyde, formic acid, phenols, acetaldehyde, and alcohols. CAT in conjunction with oxidative stress is widely represented in research for the last twenty years. For several decades, it was established that the levels of CAT relate to numerous pathologies as antioxidants in general. There was a noticeable interest in CAT in the field of transplantation in the last decade, with similar peaks as other biomarkers (Table 2).

Superoxide dismutase (SOD) is a group of enzymes functioning as the crucial part of the antioxidant defense against highly reactive superoxide radicals, partitioning them (dismutation) into H_2_O_2_ and O_2_. There are four isoenzymes, which depend on the species and intracellular localization. Those metalloproteins bind copper and zinc, manganese, iron, or nickel. They work together with glutathione peroxidase and catalase, and their activity is highly responsive to oxidative stress. Superoxide (O_2_^•−^) is produced as a by-product of oxygen metabolism. SOD catalyzes the dismutation (or partitioning) of this radical into ordinary molecular oxygen (O_2_) and H_2_O_2_. A series of reactions involves metal cations with the change of their oxidation state up to +3 to transfer and pair an electron in the superoxide. There are three forms in humans: SOD1 is located in the cytoplasm, SOD2 in the mitochondria, and SOD3 is extracellular. Despite the fact that the superoxide anion radical (O_2_^•^) spontaneously dismutes, SODs significantly speed up the mentioned reaction and outcompete damaging reactions of superoxide, protecting the cell from toxicity.

Glutathione peroxidase (GPx) is the general name of an enzyme family with peroxidase activity. It exists in two forms: selenium-dependent and selenium-independent, and it catalyzes the reduction of H_2_O_2_ or organic peroxide (ROOH) to water or alcohol [63]. The process occurs in the presence of GSH, which is converted into GSSG (oxidized glutathione) during this reaction. It is crucial to protect the polyunsaturated fatty acids located within the cell membranes from oxidative stress. Thus, GPx functions as a part of a multicomponent antioxidant defense system within the cell [64]. It is mainly represented in the kidney and the liver [62]; however, it is known for its relation to pathologies in other organs. GPx is the first enzyme that is activated under high levels of ROS. Usually, it is measured spectrophotometrically or direct assay by linking the peroxidase reaction with glutathione reductase with measurement of the conversion of NADPH to NADP. General and transplantation research interest is less expressed than CAT or SOD; however, it represents similar peaks (Table 2).

Glutathione S-transferases (GSTs) are a family of metabolic isozymes best known for their ability to catalyze the conjugation of the reduced form of glutathione (GSH) to xenobiotic substrates for detoxification. There are three forms: cytosolic, mitochondrial, and microsomal. Conjugation of GSH via a sulfhydryl group to electrophilic centers of various substrates is catalyzed by GSTs, and such compounds become more water-soluble. In addition, nucleophilic GSH reacts with electrophilic carbon, sulfur, or nitrogen atoms of nonpolar xenobiotic substrates, preventing cellular proteins, lipids, and nucleic acids from interacting with toxic, reactive substances.

Glutathione reductase (GR) catalyzes the reduction of glutathione disulfide (GSSG) to the sulfhydryl form glutathione (GSH). Thus, it prevents oxidative stress by maintaining proper cell function and GSSG/GSH ratio, while it is crucial for the cell to keep high levels of GSH. Its activity as a biomarker can be monitored by the NADPH consumption, with absorbance at 340 nm.

There are at least two more “novel” protein enzyme antioxidants: heme oxygenase 1 (HO-1) and NADPH-quinone oxidoreductase-1 (NQO1). The first one catalyzes the degradation of heme to biliverdin/bilirubin, ferrous ion, and carbon monoxide (CO). HO-1 is a member of the heat shock protein (HSP) family identified as HSP32, with the highest concentrations in the spleen, liver, and kidneys, and on the cellular level is primarily located in the endoplasmic reticulum. HO-1 is a subject of an extensive investigation into its regulatory signaling, immunomodulatory, and cryoprotective roles due to the beneficial therapeutic aspects of biliverdin and carbon monoxide [65]. HO-1 gained interest due to its antioxidant properties and role in several human diseases, including atherosclerosis, Alzheimer’s, and organ transplant rejection. It can protect against vascular remodeling and atherogenesis [66]. Bilirubin created from heme has radical-scavenging properties. HO-1 regulates a wide variety of anti-inflammatory, antioxidant, and antiapoptotic pathways. It limits heme availability for maturation of the Nox2 subunit of NADPH oxidase, prevents assembly of a functional enzyme, and reduces cellular ROS generation [67]. CO generated by HO-1 has antiproliferative, anti-inflammatory, and vasodilator properties. Anti-inflammatory and antiapoptotic effects emerge via the mitogen-activated protein kinase (MAPK) pathway [68]. The potential cytotoxic effects of iron are limited by the simultaneous enhancement of intracellular ferritin [69]. General research interest peak falls on average at the year 2010. However, there was a noticeable, relatively constant interest in HO-1 in the field of transplantation during the last 20 years (Table 2).

NQO1 performs a reduction of quinones to hydroquinones. It is a two-electron reaction, which does not result in the production of radical species, like one-electron reduction performed by, e.g., NADPH: cytochrome c oxydoreductase. Typical substrates are ubiquinone, benzoquinone, juglone, and duroquinone. Quinonoid compounds generate reactive oxygen species via redox cycling mechanisms and arylating nucleophiles. NQO1 removes a quinone from biological systems in the detoxification reaction involving NADPH, which ensures complete oxidation of the substrate without the formation of semiquinones and ROS. NQO1 plays a role in the metabolism of ubiquinone and vitamin E quinone. It protects cellular membranes from peroxidative injury in their reduced state. The induction of NQO1 is mediated through the Keap1/Nrf2/ARE signaling pathway, which promotes the expression of cytoprotective genes. NQO1 indirectly regulates p53 and p73 tumor suppressor proteins [70]. Various mechanisms and broad influence of NQO1 have gained a lot of research interest lately, with a maximum peak in the year 2020. However, transplantation research interest is less expressed.

In the group on non-protein antioxidants, the most important is glutathione (GSH). It can prevent damage to important cellular components caused by various ROS, xenobiotics, and heavy metals. It is a tripeptide and the most abundant thiol in animal cells. The primary redox couple in animal cells is reduced (GSH) and oxidized (GSSG) states. The increased GSSG-to-GSH ratio is a measure of greater cellular oxidative stress. GSH is regenerated from GSSG by GR. Glutathione binds and activates ionotropic receptors, potentially making it a neurotransmitter [71]. Direct supplementation of glutathione as an antioxidant was not successful; however, supplementation of raw nutritional materials such as cysteine and glycine were used to generate GSH. Glutathione, as the major component of TAC, regularly emerges in peer-reviewed medical journals. The number of publications was stable in the last decade and exceeded significantly other biomarkers. Transplantation research interest was proportional to the general one, comparable with SOD (Table 2).

Coenzyme Q (CoQ_10_, ubiquinone, 1,4-benzoquinone) was mentioned in the NQO1 description. Q refers to the quinone chemical group, and 10 refers to the number of isoprenyl chemical subunits in its tail. It resembles vitamins and is fat-soluble. It participates in aerobic cellular respiration and ATP generation as a component of the electron transport chain. It remains mainly in organs with the highest energy requirements: heart, liver, and kidney. It is considered an endogenously synthesized lipid-soluble antioxidant present in all membranes. During electron transport through the iron-sulfur clusters, it can only accept one electron at a time, the one which is crucial for free radicals scavenging. Biosynthesis requires at least 12 genes. CoQ_10_ may be measured in blood plasma; however, more accurate measurements can be done in cultured skin fibroblasts, muscle biopsies, and blood mononuclear cells [72].

Alpha-lipoic acid (ALA) is an organosulfur compound synthetized for aerobic metabolism. Lipoic acid is bound to proteins and works as a cofactor for at least five enzyme systems, including intermediates of the citric acid cycle, a catabolic pathway of the branched-chain amino acids, and glycine cleavage system. ALA is a direct antioxidant; however, it may also trigger antioxidant defense, enhance cellular glucose uptake, and modulate the activity of various cell-signaling molecules and transcription factors. Antioxidant activities involve (1) direct scavenging ROS and NOS; (2) regeneration of other antioxidants, while ALA is a potent reducing agent of oxidized forms of CoQ_10_, vitamin C, and GSH; (3) metal chelation and inhibiting copper- and iron-mediated oxidative damage; (4) activation of antioxidant signaling pathways via the activation of the nuclear factor E2-related factor 2 (Nrf2) by upregulation the expression of γ-GCL and other antioxidant enzymes [73]; and (5) upregulating the insulin-phosphatidylinositide-3 kinase (PI3K)-protein kinase B (PKB/Akt) signaling pathway by inhibition of nicotinamide adenine dinucleotide phosphate (NADPH) oxidase (NOX) [74]. ALA was used as an intravenous agent to treat diabetic peripheral neuropathy [75]; however, such a supplementation did not benefit patients with Alzheimer’s disease [76].

Bilirubin (BR) occurs in the catabolic pathway of heme break down, arising from aged or abnormal red blood cells. The production of biliverdin from heme is the first step, after which the enzyme biliverdin reductase (BVR) produces bilirubin from biliverdin. Bilirubin consists of an open chain tetrapyrrole and is formed by oxidative cleavage of porphyrin in heme. It is excreted after conjugation with glucuronic acid. BR has the ability to scavenge free radicals. When bilirubin acts as an antioxidant, it is oxidized to biliverdin, which is immediately reduced to bilirubin by BVR. This cycle works analogically to GSH and GSSG [77]. The absence of cellular bilirubin leads to oxidative stress [78]. BR was reported to protect the kidney, liver, heart, and gut from ischemia-reperfusion injury [79,80]. It has several immunomodulatory effects that can dampen the immune system to promote organ acceptance [81]. There was constant increasing research interest in the BR as an oxidant, peaked in 2018–2019. Significant interest in BR as an oxidant in transplantation occupies the last ten years.

Ferritin is a universal intracellular globular protein complex that stores and releases iron in a controlled way. It is the primary intracellular iron-storage protein in all living organisms, keeping iron in a soluble and non-toxic form. Free iron is toxic to cells as it acts as a catalyst in the formation of free radicals from reactive oxygen species via the Fenton reaction, producing highly damaging hydroxyl radical [82]. Binding iron in various tissue compartments is crucial for cell survival. Under steady-state conditions, the ferritin level in the blood serum correlates with total body stores of iron. Ferritin concentrations increase drastically in the presence of an infection, cancer, and oxidative stress [83]. Iron stores of the infected body are denied to the infective agent, impeding its metabolism [84]. Our research interest resembles the one regarding bilirubin.

### 4.2. Lipid Peroxidation (LPO)

Lipids can be oxidized, chlorinated, and nitrated by a range of RS, excluding H_2_O_2_, NO^•^, or O_2_^•−^, which are unreactive with lipids. Lipid peroxidation is a complex process, and a wide range of products is formed in variable amounts.

It is the most widely known biological free radical chain (FRC) reaction. The oxidation of unsaturated fatty acids or other lipids results in products, which are peroxides of these compounds. Note that ROS does not initiate peroxidation. Their presence only intensifies the process. Peroxidation reaction, like every FRC, can be divided into three stages:

1. Initiation: Creation of fatty acid radical. ROS which initiates this reaction in living cells are hydroxyl (HO·), peroxy (LOO·), alkoxy (LO·), and alkyl (L·), as well as O_3_, SO_2_, and NO_2_. The separation of hydrogen leads to the formation of an alkyl radical. LH →L^•^ + H_2_O.

2. Prolongation: Unstable fatty acid radicals easily react with molecular oxygen (O_2_), forming peroxides, which are also unstable and react with more fatty acid molecules, creating more radicals. The reaction runs in a cycle: L^•^ + O_2_→LOO^•^ LOO^•^ + LH →LOOH + L^•^.

3. Termination: Growing number of free radicals increases the probability of collision between them, which ends the process: L^•^ + L^•^ →L–L, LOO^•^ + LOO^•^ →L=O + LOH + O_2_, LOO^•^ + L^•^ →L=O + LOH. Termination reaction results in products such as dimers of fatty acids, hydroxy acids, and oxoacids.

Lipid peroxidation products often react with proteins present in the cell membranes, creating protein–lipid adducts. Further reactions of lipid peroxidation products lead to aldehydes syntheses such as MDA or 4-hydroxynonenal. They easily diffuse through biological membranes and can be the indirect cause of DNA damage by ROS. They are cytotoxic, mutagenic, and carcinogenic and can cause a rupture in DNA strands. Lipid peroxidation affects all cell membranes leading to their damage and loss of function. The process can impair ion pumps or electron transport in the respiratory with subsequent diminished ATP production [85].

The basic substances for evaluating lipid peroxidation are thiobarbituric acid reactive substances (TBARS) and malondialdehyde (MDA). TBA-reactive material is questioned in modern evaluation because most of it in human body fluids is reported not to be related to lipid peroxidation; however, high-performance liquid chromatography (HPLC) can provide accurate measurements of MDA-TBA chromogen to assay MDA directly. MDA is more specific for lipid peroxidation, but only one of many aldehydes formed during such a reaction, and MDA can also arise from the free radical attack on sialic acid and deoxyribose [86]. MDA is also a biomarker far from perfect because some MDA–amino acid adducts that can be absorbed through the gut thus can be confounded by diet [86]. DNA adducts are formed by the reaction of MDA with deoxyadenosine and deoxyguanosine in DNA. The primary mutagenic one is M1G, which adds oxidative damage to the cell [87]. The assessment of the membrane damage in spermatozoa and the evaluation of sperm function by altering membrane fluidity, permeability, and impairing sperm functional competence can be done using MDA levels [88]. There were more modern markers reported, such as 4-hydroxynonenal and acrolein, but the substances which fit the best the criteria of lipid peroxidation biomarkers are isoprostanes, specific end products of the peroxidation of polyunsaturated fatty acids, with the most represented group—F_2_-isoprostanes, which arise from arachidonic acid [89]. Isoprostanes can be derived from eicosapentaenoic acids or docosahexaenoic acids forming F_4_-isoprostanes, also called neuroprostanes, due to their contribution to neurodegenerative diseases [90]. Commercially available immunoassay kits can easily measure them and plasma as in the urine, although they also do not meet the criteria for ideal biomarkers because their concentration may be influenced by O_2_ concentration and metabolic rate. It could be crucial for evaluating ischemia-reperfusion injury in transplantation due to both factors’ impairment during the process. Isoprostanes were specially studied in cerebrospinal fluid in Alzheimer’s disease [91]. Isoprostanes were reported to be influenced by oral antioxidants such a vitamin C or E [92,93], as an example on a much broader discussion about the possible therapies to oxidative stress; however, there were also claims that isoprostane levels vary with time of day and from day to day as oxidative stress itself [94].

ACR, CRA, and 4-HNE are α,β-unsaturated aldehydes, the most reactive and toxic ones. Thus, these lipid peroxidation products can modify nucleophilic side chains on amino acids: sulfhydryl groups of cysteine, imidazole groups of histidine, and the amino acid groups of lysine [95]. The formation of adducts by these aldehydes has been linked to the disruption of cell signaling and mitochondrial dysfunction [96]. α,β-Unsaturated aldehyde-induced toxicity is reported to occur because of depletion of cellular GSH, which subsequently induces ROS production that leads to cell malfunction [97]. NF-κB activation occurs directly by binding to the reactive cysteine on the subunit of IκB kinase (IKK) or indirectly by decreasing cellular GSH [98].

ACR has a strong electrophilic reactivity towards nucleophiles; therefore, it disrupts the redox control of protein function and causes cytotoxicity via irreversible adduction. It is an important oxidative stress biomarker for LPO. ACR levels increase with aging and in certain diseases involving oxidative stress and inflammation such as atherosclerosis and Alzheimer’s disease [99,100], atherosclerosis [101], hypertension [102], dyslipidemia [103], and infarction [104].

CRA is produced during the combustion of carbon-containing compounds, including cigarette smoke; however, intracellularly, it is formed as a product of LPO. CRA reduces levels of GSH due to enzymatic conjugation and it is mutagenic [105]. CRA modulates biological reactions through various related signaling pathways increasing cellular oxidative stress [106].

4-HNE is a potent alkylating agent that reacts with DNA and proteins, generating various adducts, which can induce stress signaling pathways and apoptosis. Consequently, 4-HNE is associated with several pathological conditions, such as COPD [107], acute respiratory distress syndrome (ARDS) [108], and atherosclerosis [109].

7-Ketocholesterol (7KC) is a toxic product of a reaction between cholesterol and ROS. 7KC accumulation can cause significant damage to membranes, signaling pathways, and overall cell function. 7KC is the most prevalent nonenzymatically produced oxysterol in-vivo [110]. In addition, 7KC increases ROS production by activation of NADPH oxidase (NOX) and triggers an apoptotic stress response [111]. 7KC has been linked to poly-ADP-ribose (PARP) formation, which has implications in many age-related diseases, including cancer, atherosclerosis, and Alzheimer’s disease [112,113]. Final 7KC-induced cell death is caused by interference with biological processes, including enzymatic reactions, membranes functions, molecules oxidation, and apoptosis induction.

In Table 3, we summarized biomarkers of lipid peroxidation (BLP) with histograms of research interest. General interest was determined by search formula: {(“biomarker”[Title/Abstract]) AND (oxidative stress)}. Research interest in the field of transplantation, as a Medical Subject Headings, was determined by search formula: {(“biomarker”[Title/Abstract]) AND (transplantation)}. In the last decade, there was a significant growing interest in MDA, with proportional interest in transplantation. However much less expressed, two other “popular” were TBARS and 4-HNE with similar reflection in transplantation. Surprisingly there was a little interest in isoprostanes in transplantation.

### 4.3. Protein Oxidation

The inherent effect of aerobic cellular metabolism is the oxidation of biomolecules and protein degradation. Oxidative damage to proteins bears severe consequences because it affects the function of receptors, enzymes, and transport proteins and contributes to indirect damage to other biomolecules, e.g., DNA repair enzymes or polymerases in DNA replication. Oxidation of amino acid residues, such as tyrosine, leads to the formation of dityrosine, protein aggregation, cross-linking, and fragmentation. The free radical attack on proteins generates amino acid radicals, which may crosslink or react with O_2_, resulting in peroxyl radicals’ generation, which may further turn into protein peroxides by abstracting (H^•^) and triggering more free radicals. The loss of activity and inactivation of mentioned proteins impairs various chemical reactions and metabolic pathways, ending in cell death [31]. There were different individual amino acids oxidation products measured in humans: kynurenines (from tryptophan), bityrosine, valine, and leucine hydroxides, L-dihydroxyphenylalanine (L-DOPA), ortho-tyrosine, 2-oxo-histidine, glutamate semialdehyde, and adipic semialdehyde. Advanced oxidation protein products (AOPP) are bityrosine-containing protein cross-linking products. Advanced glycation end-products (AGE) are protein carbonyl compounds produced by protein-ROS interaction. They can be detected in the plasma of dialyzed patients [115].

During oxidative stress, mainly the thiol groups (-SH) undergo the oxidation reaction. It can be initiated by ROS such as O_2_^•^^−^, H_2_O_2_, or HO^•^ resulting in the formation of thiol radicals (RS^•^), which are immediately dimerized to sulfides: RSH + O_2_^•−^ + H^+^ → RS^•^ + H_2_O_2_, 2RSH + H_2_O_2_→ RS^•^ + 2H_2_O, RSH + HO^•^→ RS^•^ + H_2_O, 2RS^•^→ RSSR. The oxidation of polypeptide chains is a similar process to lipid peroxidation. Therefore, it regards polypeptide chain or amino acid residues directly, with aromatic ones as the most reactive. When α-amino acid carbon releases a proton when interacting with hydroxyl radicals, it creates an alkyl radical converted to alkyl hydroperoxide by reaction with oxygen. The alkyl radical can be converted to an alkoxy radical, which activates the fragmentation of the polypeptide chain by thiol damage. It leads to the mentioned protein function loss [85].

Tyrosine is one of the major targets of protein oxidation. Dityrosine, formed by the free radical attack on a wide range of proteins, is easily detectable in urine, and its increased excretion was reported in sepsis and kwashiorkor [116]. It can be detected by an immunohistochemical method with the use of an acetyltyramine–fluorescein probe to detect the formation of tyrosine radicals in proteins in cells subjected to oxidative stress. The highly reactive hydroxyl radical oxidizes phenylalanine residues to o-tyrosine and m-tyrosine [117]. Proteins can also be attacked by reactive chlorine, bromine, and nitrogen species resulting in the formation of 3-chlorotyrosine, para-hydroxyphenylacetaldehyde, 3,5-dichlorotyrosine, 3-bromotyrosine, and 3-nitrotyrosine. Those products are usually measured by immunostaining, HPLC, and MS; however, methods have some limitations or artifacts. HPLC lacks accuracy since some metabolites arise by other mechanisms, body fluids, or tissues exposed to acid or freezing can cause artefactual nitration of tyrosine, nitrotyrosine can be destroyed by hypochlorous acid, generated by activated neutrophils, immunostaining is prone to artifacts. MS-based assays can detect nitro-, chloro-, and ortho-tyrosines. There are proteomics methods, which can separate proteins for the recognition of modified ones. There is also a possibility to assess protein nitrosylation, which is a reversible modification involving the attachment of NO^•^ to a metal site or a cysteine residue.

The method most frequently used for the assessment of protein damage is the carbonyl assay. Carbonyls arise due to protein glycation by sugars, by the binding of aldehydes (including those formed by LPO) to proteins, and by the direct oxidation of amino-acid side chains by RS to generate such products as glutamate or aminoadipic semialdehydes. Carbonyl groups (CO) are measured using derivatization of the carbonyl group with 2,4-dinitrophenylhydrazine (DNPH), which leads to the formation of a stable dinitrophenyl (DNP) hydrazone product. The product can be detected using enzyme-linked immunosorbent assay (ELISA), spectrophotometric assay, and one-dimensional or two-dimensional electrophoresis followed by Western blot immunoassay [118]. Samples can be derived from blood plasma as well as other body fluids and tissues. Carbonyls are not specific markers of oxidative damage because mentioned methods also measure bound aldehydes and glycated proteins. The carbonyl assay is a quantitative method allowing to assess the average extent of protein modification. For more specific identification, proteomic techniques are used [31]. Research interest in PubMed database was determined by search formula: {(“biomarker”[Title/Abstract]) AND (oxidative stress/transplantation)}. There is an increasing number of publications about DT, however not in the field of transplantation. There was a significant research interest in NY and protein carbonyls in the last twenty years, peaked in 2013. It was relatively represented in the field of transplantation. Biomarkers of proteins peroxidation were presented in Table 4.

### 4.4. Nucleic Acid Oxidation

Oxidative DNA damage, mostly due to the hydroxyl radical, generates a huge range of base and sugar modification products [119]. They can be measured by HPLC, gas chromatography–mass spectrometry (GC–MS), liquid chromatography–mass spectrometry (LC–MS), and antibody-based techniques, but none of these have been established as a gold standard [31]. It is estimated that each cell under normal conditions is the target of several thousand attacks on its DNA every day. Free radicals attacking DNA purines, pyrimidines, and deoxyribose create initial products, which undergo further transformations. Damage to DNA by free radicals occurs much less frequently than oxygen damage to proteins and lipids. However, the consequences are more serious due to mutagenic or immunogenic changes. One of the major products of DNA oxidation is 8-hydroxy-2′-deoxyguanosine (8-OHdG) and 8-oxoguanine (8-oxo-Gua) [120]. Guanine, an aromatic heterocyclic compound and purine (systematic name is 2-amino-6-hydroxypurine), is a basic building block of both DNA and RNA forming a complementary pair with cytosine, both in the free state and as a nucleoside, is particularly susceptible to the effects of free radicals in the C8 position. Thus, 8OHdG is the most favorable best-known mutagenic modification of DNA [85]. 8-OHdG/dG (deoxyguanosine, one hydroxyl group removed from the 2′ position of the ribose sugar) ratio were determined as reliable oxidative stress markers [121]. All the repair products can be measured by gas chromatography-mass spectrometry (GC/MS), however typical methods for the assessment for 8-oxodG and 8-oxoGua is high-performance liquid chromatography (HPLC) including electrochemical detection (EC) and ultraviolet (UV) absorbance detection for thymidine glycol (dTg) and thymine glycol (Tg); however, none of them became standard. It may arise from at least a couple of problems: artefactual 8OHdG oxidation during DNA isolation, differences in the oxidation of nuclear and mitochondrial DNA, external antioxidants activity including diet, or localization of damage—active or inactive genes, telomeres or DNA without any meaning, type of cells or tissues used for samples collection. It is possible to evaluate 8OHdG in urine, however it is also prone to artifacts: 8OHdG can arise from degradation of oxidized dGTP in the DNA precursor pool and DNA in foods can be oxidized during storage and cooking, so confounding by diet cannot yet be ruled out—oxidized bases can be absorbed and re-excreted. Additionally, the problem with all these assays is the separation of the very small amounts of analyte from urine [122]. Research interest in PubMed database was determined by search formula: {(“biomarker”[Title/Abstract]) AND (oxidative stress/transplantation)}. We can observe growing interest in 8OHdG for the last three decades, with an emerging plateau for the last six years, with moderate interest on the field of transplantation and the peak in 2019. The interest in 8-oxo-Gua was much smaller and quite constant. Biomarkers of nucleic acids peroxidation were presented in Table 5.

## 5. Regulatory Pathways and Antioxidative Therapies

Despite severe impairment to the cell structure, oxidative stress is also the cause of significant energy depletion and metabolic resources. Oxidative stress reduces ATP concentration in the cells, which is caused by damage to mitochondria and by deactivation of glyceraldehyde-3-phosphate dehydrogenase, which inhibits glycolysis reactions. Oxidative stress and increased GSSG-to-GSH ratio result in increased consumption of ATP and increased catabolism of adenine nucleotides. Xenobiotics and the products of lipid and protein peroxidation are disposed of outside the cell after conjugation with GSH with additional ATP usage (also due to active transport of GSSG) and GSH depletion. Deactivation of the calcium pump leads to an increase in calcium ion concentration in the cytoplasm. Deactivation of K, Ca, and Na channels increases cell membrane permeability and depolarization of the cell membrane [85]. It was reported that there is a relation between the inactivation mechanisms of the antioxidant system and the generation of ROS at a physiological level. In high oxidative stress conditions, the endogenous antioxidant defenses would not be enough. The equilibrium of the ROS antioxidant defense mechanisms requires oxidative modification in the cellular membrane or intracellular molecules and signaling pathways adjusting the cell to the actual needs [123]. Basic activation paths were presented in Figure 3. Connecting lines were drawn according to existing relations confirmed by available research from the last 20 years. Their thickness is proportional to the research interest measured by the number of publications found on a particular subject.

One of the most represented pathways was PI3K/Akt/Nrf2 activation. Nuclear factor-erythroid 2-related factor 2 (Nrf2) is a pivotal transcription factor that maintains cellular redox homeostasis. When oxidative stress occurs, Nrf2 dissociates from its inhibitor Kelch-like ECH-associated protein 1 (Keap1) and binds to the antioxidant response element (ARE) to promote the gene expression of detoxifying enzymes and antioxidant enzymes [124]. Potential drugs interacting with PI3K/Akt/Nrf2: 6′-O-galloylpaeoniflorin (GPF) [125], vasicine [126], phosphocreatine [127], resveratrol [128], etc.

The mTOR (mammalian target of rapamycin) is a protein serine/threonine kinase and a component of PI3K, which regulates cell growth, proliferation, motility, and survival as well as gene transcription and protein synthesis that are activated in response to hormones, growth factors, and nutrients. mTOR consists of two complexes mTORC1 and mTORC2, activated by Akt subsequently phosphorylated by PI3K [129]. Numerous studies link mTOR and various tumor types, chronic inflammation, and oxidative stress [130]. Potential drugs interacting with PI3K/Akt/mTOR: mangiferin [131], atractylenolide III [132], and epigallocatechin gallate (EGCG) [133].

NF-κB is a protein complex that controls transcription of DNA, cytokine production, and cell survival. It responds to stimuli such as stress, cytokines, free radicals, heavy metals, ultraviolet irradiation, oxidized LDL, and bacterial or viral antigens. It plays a key role in regulating the immune response to infection, cancer development, and inflammatory and autoimmune diseases [134]. NF-κB belongs to the category of “rapid-acting” primary transcription factors, making it the first responder to harmful cellular stimuli. Activation of the NF-κB is initiated by the IκB kinase (IKK). Degradation of IκB allows the freed NF-κB complex to enter the nucleus where it can “turn on” the expression of specific genes, which play a major role in regulating the amount of ROS in the cell [135]. Potential drugs interacting with PI3K/Akt/NF-κB: rigosertib [136], methane [137], glycyrrhizin (GLY)-triterpenoid saponin glycoside [138].

The forkhead (Fox) transcription factor O is a family of proteins (FoxO). They are involved in various biological processes: metabolism, cell proliferation, apoptosis, and oxidative stress response [139]. Subtypes include FoxO1, FoxO3, FoxO4, and FoxO6. FoxO4, initially recognized as a tumor suppressor, was linked with diabetic nephropathy [140], diabetic retinopathy [141], and ischemic limbs [142]. FoxO4 activation is also presents in ROS-induced cell apoptosis [143]. Potential drugs interacting with FoxO: plant iridoids [144], glochidion zeylanicum leaf extracts [145], and coenzyme Q_10_ [146].

The antiapoptotic effects of glutathione S-transferase P1 (GSTP1) are worth mentioning. They might be mediated through interaction with c-Jun NH(2)-terminal kinase (JNK). The research interest in GSTP1 covers last 22 years with a peak in 2012. GSTP1 widely represented in oncology connecting oxidative stress, apoptosis and carcinogenesis [147,148,149], but also plays a role in asthma [150], Parkinson’s disease [151]. Kidney damage was also reported in conjunction, however much less supported [151].

There is a growing research interest in cellular signaling pathways since it was proven that they are a nexus of various regulatory mechanisms. In addition to those mentioned above, there are many more reported: JAK/STAT, HMGB1, HO-1, TLR4, JNK, ERK1, MAPK, Omi, HtrA2, AMPK, SIRT3, NLRP3, AGE/RAGE. They create certain pathways, usually involving a couple of them. Recently reported: JAK2/STAT3-CPT1a, Akt/GSK-3 β/Nrf2, SIRT1/HMGB1/NF-κB, Akt/Nrf2/HO-1, ROS/JNK/p3, TLR4/MyD88/NF-κB, c-Raf-Erk1/2-p90 rsk-CREB, IL-10/GSK-3β/PTEN, MAPK/Nrf2/HO-1, etc. There are numerous drugs or complex compounds reported to interact with cell signaling. Detailed presentation of pathways and therapeutics exceeds the subject of this review; however, there are recently reported compounds worth mentioning: melatonin, thiamine, sildenafil, taurine, kidney extracellular matrix hydrogel, water-soluble coenzyme Q10, C1q/TNF-related protein 6 (CTRP6), 20-hydroxyeicosatetraenoic acid (20-HETE), vaporized perfluorocarbon, bilirubin, N-acetylcysteine, hydrogen gas, propofol, and many others.

## 6. Conclusions

Oxidative stress is a complex phenomenon resulting from the imbalance of cell homeostasis, leading to oxidative damage. It plays a vital role in the pathogenesis of many diseases. Oxidative stress is directly caused by reactive species, mainly ROS. Some of them contain unpaired electrons—free radicals, and others do not, but all of them are very reactive and cause the peroxidation of various molecules. Damage to so many cellular structures can result in the deterioration of function, including apoptosis and necrosis. Oxidative damage leaves a mark, which can be detectible by specific methodology regarding affected molecules. Those substances become the biomarkers of oxidative stress due to their ability to represent the intensity of biochemical changes accurately. The group of biomarkers, in general, comprises internal compounds related to cell physiology as well as end products of peroxidation reactions. Maintaining the cell homeostasis and redox state is a matter of competing chemical reactions and a very complex cellular and nuclear signaling mechanism. The cell can react to the environment and redistribute the energy and resources to overcome certain adverse events as oxidative stress. Oxidative damage, stress, and ROS are still intensively exploited research subjects, especially kidney disease and renal transplantation. However, the modern approach regards mainly signaling pathways and cell internal regulation; therefore, further research is necessary to translate the biochemical correlations into clinical practice and organ protection. With time, adequate interventions and solutions at the cellular level will lead to outcome improvement after organ transplantation.

## Figures and Tables

**Figure 1 ijms-22-08010-f001:**
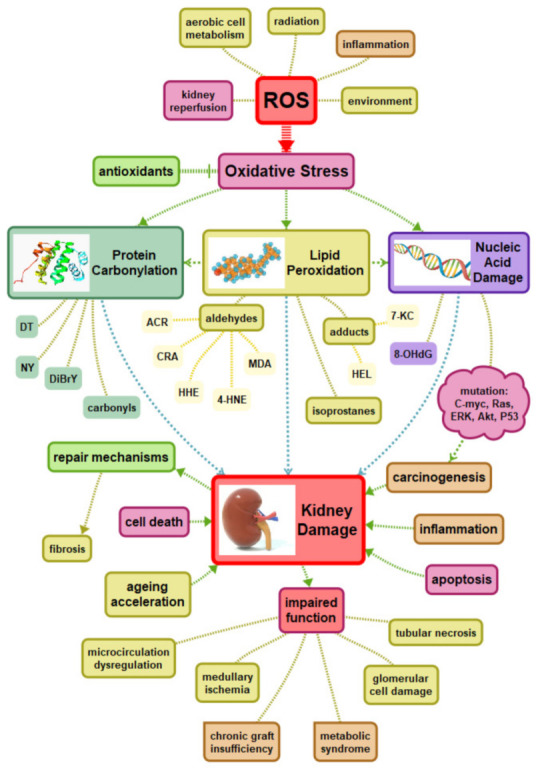
Oxidative stress and kidney damage. DT—tyrosine, NY—3-nitrotyrosine, DiBrY—dibromotyrosine, ACR—acrolein, CRA—crotonaldehyde, HHE—4-hydroxy-trans-2-hexenal, 4-HNE—4-hydroxynonenal, 7-KC—7-ketocholesterol, HEL—hexanoyl-lysine adduct, 8OHdG—8-hydroxy-2′-deoxyguanosine.

**Figure 2 ijms-22-08010-f002:**
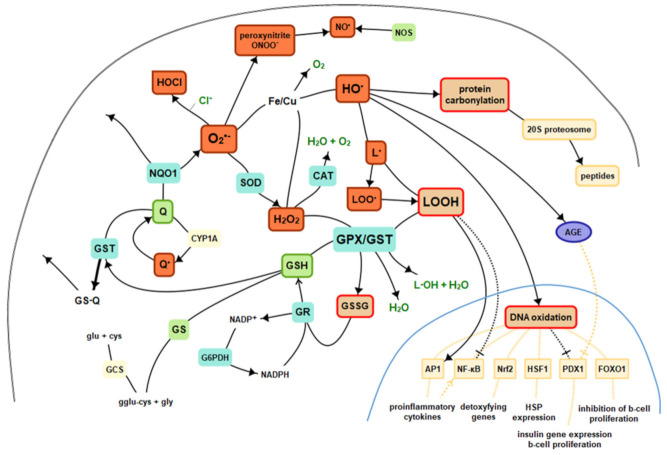
Basics of intracellular antioxidant mechanisms and nuclear signaling. Legend: CAT—catalase, GPx—glutathione peroxidase, GR—glutathione reductase, GST—glutathione S-transferases, GSH—reduced glutathione, GSSG—oxidized glutathione, Q—coenzyme Q_10_, GS—glutathione synthetase, HO·—hydroxyl radical, H_2_O_2_—hydrogen peroxide, L·—alkyl radical, LO·—alkoxy radical, LOO·—peroxy radical, HOCl—hypochlorous acid, NO—nitric oxide, NOS—nitric oxide synthase, NQO1—NADPH-quinone oxidoreductase-1, O_2_—molecular oxygen_,_ O_2_^•—^superoxide, SOD—superoxide dismutase, AGE—advanced glycation end products, NF-κB—nuclear factor kappa-light-chain-enhancer of activated B cells, AP1—activator protein 1, Nrf2—nuclear factor-erythroid 2-related factor 2, HSF1—heat shock factor 1, PDX1—insulin promoter factor 1, FoxO—forkhead transcription factor O.

**Figure 3 ijms-22-08010-f003:**
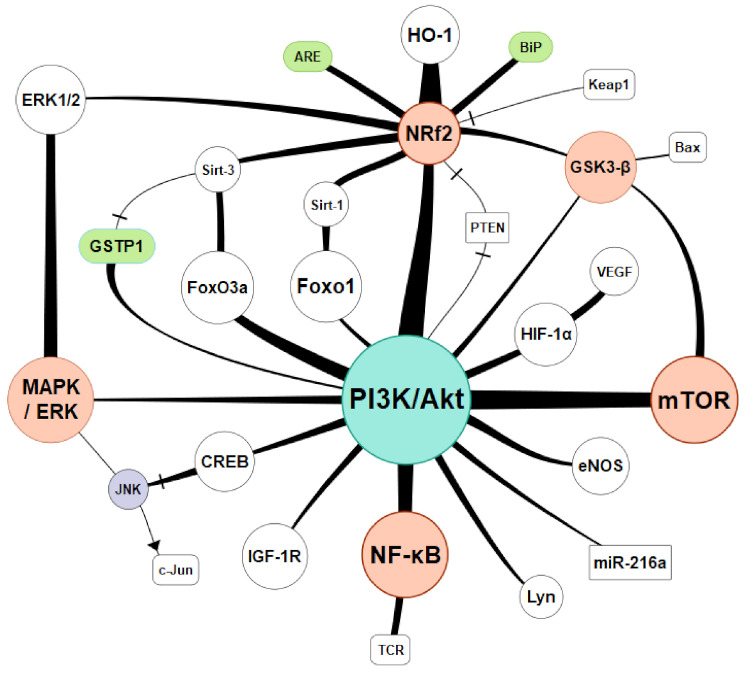
Basics of intracellular antioxidant mechanisms and nuclear signaling. Legend: Akt—serine/threonine-specific protein kinase B (PKB), ARE—antioxidant response element, Bax—apoptosis regulator known as bcl-2-like protein 4, BiP—binding immunoglobulin protein (GRP-78), CREB—cAMP response element-binding protein, cellular transcription factor, c-Jun protein that in combination with c-Fos forms the AP-1 early response transcription factor, eNOS—endothelial NOS, GSK-3β—glycogen synthase kinase 3 beta, HIF-1α—hypoxia-inducible factor 1-alpha, GSTP1—glutathione S-transferase pi 1, HO-1—heme oxygenase 1, IGF-1R—insulin-like growth factor 1 receptor, Keap1—Kelch-like ECH-associated protein 1, Lyn—Tyrosine-protein kinase Lyn, MAPK/ERK—Ras-Raf-MEK-ERK pathway-mitogen-activated protein kinases, originally called ERK, extracellular signal-regulated kinases, JNK—c-Jun N-terminal kinase, miR-216a—microRNA 216a tumor suppressor regulating the cell proliferation, mTOR—mammalian target of rapamycin, NF-κB—nuclear factor kappa-light-chain-enhancer of activated B cells, PI3K—phosphatidylinositol 3-kinase, PTEN—tumor suppressor gene (phosphatase and tensin homolog), SIRT—Silent Information Regulator, TCR—T-cell receptor inflammatory signaling, VEGF—vascular endothelial growth factor, -/-—inhibition.

**Table 1 ijms-22-08010-t001:** Reactive oxygen species—description and research interest.

ROS	Description	General	Transplantation
O_2_^•−^ superoxide anion	one-electron reduction state of O_2_, formed in autoxidation reactions and by the electron transport chain, can release Fe^2+^ from iron-sulfur proteins and ferritin, undergoes dismutation to H_2_O_2_ spontaneously or by enzymatic catalysis, the precursor for metal-catalyzed OH^•^ formation, detectable as a biomarker	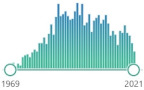 1843	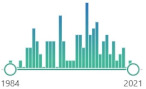 102
OH^•^ hydroxyl radical	three-electron reduction state, formed by Fenton reaction and decomposition of peroxynitrite; extremely reactive, attack most of the cellular components	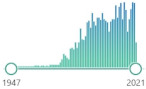 2154	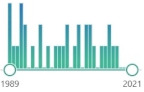 30
RO^•^, alkoxyl ROO^•^, peroxyl radicals	oxygen centered organic radicals, lipid forms participate in lipid peroxidation reactions, produced in the presence of oxygen by radical addition to double bonds or hydrogen abstraction	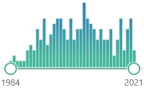 203	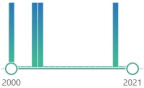 4
H_2_O_2_ hydrogen peroxide	two-electron reduction state, formed by dismutation of O_2_^•−^ or by direct reduction of O_2_; stable, lipid-soluble, able to diffuse across membranes	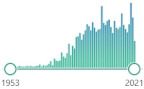 2073	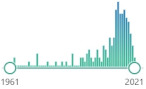 155
^1^O_2_ singlet oxygen	systematically dioxidene, highly reactive toward organic compounds (more than O_2_) and unstable, responsible for the photodegradation of many materials, pollutes urban atmospheres	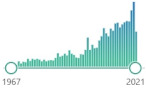 2226	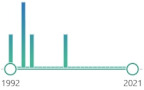 5
O_3_ ozone	allotrope of oxygen, colorless or pale blue gas, a powerful oxidant, damage mucous and respiratory tissues, harmful to people at levels currently found in urban areas, affects the respiratory, cardiovascular, and central nervous system	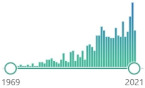 582	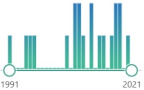 19
ROOH organic hydroperoxide	stable, formed by radical reactions with cellular components such as lipids and nucleobases	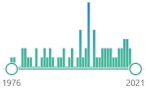 64	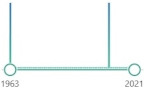 2
HOCl hypochlorous acid	formed from H_2_O_2_ by myeloperoxidase, lipid-soluble and highly reactive, oxidizes protein constituents, including thiol groups, amino groups, and methionine	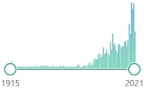 744	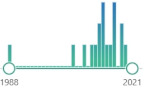 14
HOBr hypobromous acid	weak, unstable acid, “reactive bromine species”; generated biologically as a disinfectant, especially by eosinophils; an oxidizer; especially effective when used in combination with its congener, hypochlorous acid	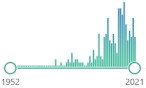 233	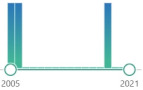 2
NO^•^ nitric oxide	colorless gas; free radical; signaling molecule; “Molecule of the Year” in 1992; 1998 Nobel Prize in Physiology or Medicine; endothelium-derived relaxing factor (EDRF); converted to nitrates and nitrites by oxygen and water	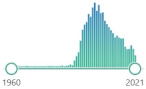 3962	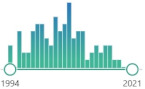 81
NO_2_^•^ nitrogen dioxide	reddish-brown gas; good oxidizer; exposed and are at risk for occupational lung diseases; chemically reacts with antioxidant and lipid molecules; classified as an extremely hazardous substance	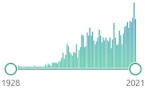 1544	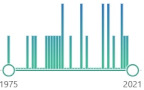 24
ONOO^–^ peroxynitrite	formed in a rapid reaction between O_2_^•−^ and NO^•^, lipid-soluble and similar in reactivity to HOCl, protonation forms peroxynitrous acid, which can undergo homolytic cleavage to form hydroxyl radical and nitrogen dioxide	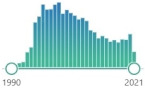 2722	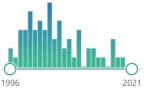 62

**Table 2 ijms-22-08010-t002:** Endogenous antioxidants—description and research interest.

Biomarker	Description	General	Transplantation
CAT catalase	tetramer, enzyme, biomarker, catalyzes the decomposition of hydrogen peroxide to water and oxygen, has one of the highest turnover numbers of all enzymes, first noticed in 1818	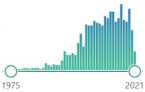 1068	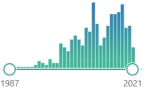 356
SOD superoxide dismutase	enzyme, biomarker, metalloprotein connected in humans with Zn/Cu or Mn, discovered in 1968	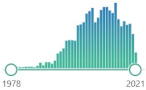 2117	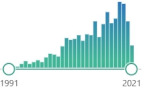 715
GPx glutathione peroxidase	selenium-containing enzyme family, several isozymes are encoded by different genes, biomarker, the protective system depends heavily on the presence of selenium, discovered in 1957	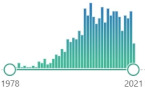 763	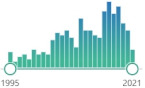 296
GST glutathione S-transferase	family of metabolic isoenzymes, biomarker, catalyze the conjugation of the reduced form of glutathione (GSH) to xenobiotic substrates, nomenclature first proposed in 1992	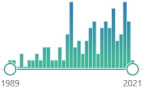 104	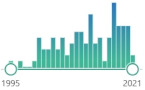 95
GR glutathione reductase	homodimer disulfide oxidoreductase enzyme encoded by the GSR gene, biomarker, catalyzes GSSH and regenerates it to GSH, involves NADPH and FAD binding domains, first purified in 1955	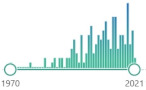 159	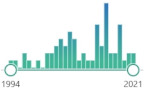 63
HO-1 heme oxygenase 1	an enzyme, mediates the first step of heme catabolism, cleaves heme to biliverdin, carbon monoxide, and ferrous iron, encoded by HMOX1 gene induced in oxidative stress and inflammation, first characterized in 1962	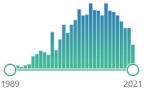 1457	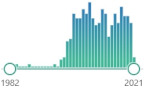 404
NQO1 NADPH-quinone oxidoreductase-1	protein homodimer, detoxifying enzyme, present in cytosol, performs a two-electron reduction of quinones to hydroquinones and of other redox dyes without the formation of ROS,	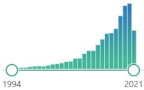 1837	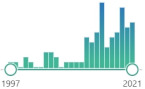 86
GSH glutathione	L-γ-glutamyl-L-cysteinyl-glycine, non-enzymatic, biomarker, reduced form (GSH) and glutathione disulfide (GSSG)—primary redox couple in animal cells	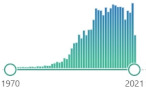 5118	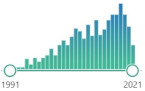 798
coenzyme Q ubiquinone	benzoquinone derivate, localized in the mitochondrial respiratory chain and other internal membranes, coenzyme, first isolate was in 1950 in the lining of a horse’s gut	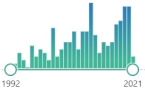 185	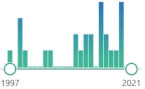 22
ALA α-lipoic acid	delivers antioxidant activity in nonpolar and polar mediums, effective in recharging enzymes in the mitochondria, might counteract NF-κB activation	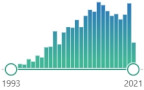 1430	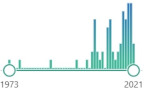 62
BR bilirubin	product of heme degradation works as an antioxidant in cycle BR-biliverdin, relevantly documented in 1827	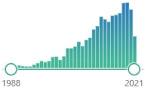 1941	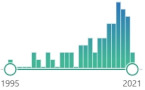 64
ferritin	intracellular globular iron-binding protein, small amounts are secreted into the serum, prevents ROS generation via the Fenton reaction by binding iron	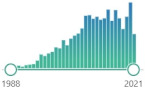 1678	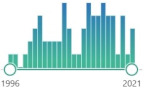 60

**Table 3 ijms-22-08010-t003:** Biomarkers of lipid peroxidation—description and research interest.

Biomarker	Description	General	Transplantation
TBARS TBA-reactive substances	the oldest and one of the most widely used nonspecific by-products of lipid peroxidation, reacts with thiobarbituric acid (TBA), forming a pink chromogen (TBARS) measured at 532–535 nm	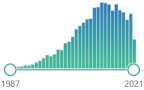 7431	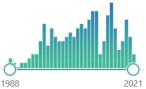 200
MDA malondialdehyde	CH_2_(CHO)_2_, colorless liquid, highly reactive, a product of LPO of polyunsaturated fatty acids, form covalent protein adducts referred to as advanced lipoxidation end-products (ALE), in analogy to advanced glycation end-products (AGE)	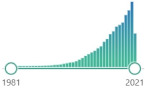 30,269	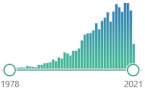 2808
4-HNE 4-hydroxynonenal	α,β-unsaturated hydroxyalkenal, produced by lipid peroxidation (arachidonic or linoleic groups) in cells in higher quantities during oxidative stress, possible role in cell signal transduction, first reported in 1991 [114], they can also come from omega-3 fatty acids	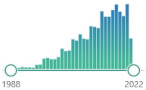 2418	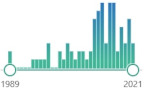 73
ACR acrolein (propenal)	the simplest unsaturated aldehyde, named and characterized in 1839, electrophilic, reactive and toxic, contact herbicide to weeds, present in tobacco smoke increases the risk of cancer, produced during cyclophosphamide treatment	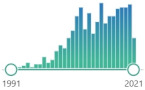 547	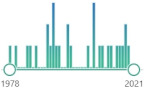 29
F_2_-isoprostanes	prostaglandin-like compounds formed in vivo from the free radical-catalyzed peroxidation of arachidonic acid, discovered in 1990,	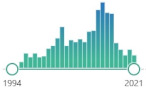 623	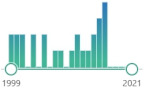 22
F_4_-isoprostanes	prostaglandin-like compounds formed in vivo from the free radical-catalyzed peroxidation of docosahexaenoic acid, potent biological activity as anti-inflammatory mediators	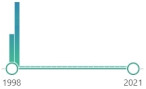 3	0
CRA crotonaldehyde	CH_3_CH, representative carcinogenic aldehyde formed endogenously through lipid peroxidation, CRA is a highly reactive aldehyde and reacts with a lysine residue in the protein, reaction with CRA and lysine residue leads to the formation of numerous numbers of adducts	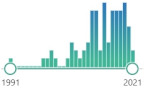 74	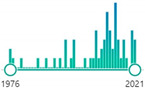 53
HHE 4-hydroxy-trans-2-hexenal	oxygenated α,β-unsaturated aldehyde, other coming from omega-3 fatty acids: 4-oxo-trans-2-nonenal, 4-hydroperoxy-trans-2-nonenal, and 4,5-epoxy-trans-2-decenal	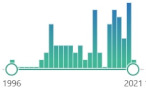 64	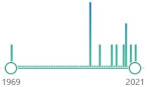 11
7KC 7-ketocholesterol (7-oxocholesterol)	toxic oxysterol, produced from oxidized cholesterol, induces: NOX, pro-inflammatory cytokines and TNF-α	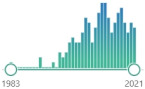 162	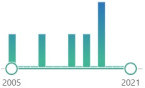 6

**Table 4 ijms-22-08010-t004:** Biomarkers of proteins peroxidation—description and research interest.

Biomarker	Description	General	Transplantation
DiBrY dibromotyrosine	product of the reaction of hypobromous acid (HOBr) from hydrogen peroxide (H_2_O_2_) and bromide ion (Br^–^), detected by anti-dibromo-tyrosine [DiBrY], mAb (3A5) JAI-MBY-020P	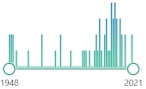 52	5 1951, 1989, 1995, 1999, 2018
DiY/DT dityrosine (bityrosine)	biphenyl compound comprising two tyrosine residues linked at carbon-3 of their benzene rings	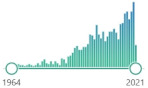 1014	6 2001, 2005, 2010, 2012, 2020(2)
m-Tyrosine o-Tyrosine	abnormal tyrosine isomers, derive from oxidation of the benzyl ring of the phenylalanine by hydroxyl radical, adversely affect cells and tissues	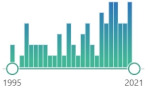 62	4 2003, 2007, 2008, 2012
NY 3-nitrotyrosine	specific marker of attack of peroxynitrite (ONOO^–^) upon proteins, measured by immunostaining, HPLC, and MS in human tissues	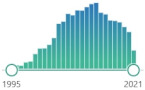 3554	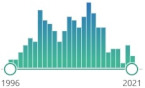 201
protein carbonyls	measurement of protein CO groups after their derivatization with DNPH is the most widely utilized measure of protein oxidation	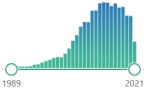 9122	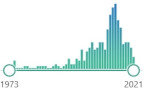 325

**Table 5 ijms-22-08010-t005:** Biomarkers of nucleic acids peroxidation—description and research interest.

Biomarker	Description	General	Transplantation
8OHdG 8-hydroxy-2′ -deoxyguanosine	oxidized derivative of deoxyguanosine, major products of DNA oxidation, increased levels are found during carcinogenesis, increases with age, linked to the enzyme OGG1 and transcription factor NFκB	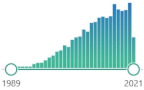 5189	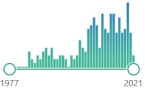 212
8-oxo-Gua 8-hydroxyguanine	one of the most common DNA lesions resulting from reactive oxygen species, modifying guanine, and can result in a mismatched pairing with adenine resulting in G to T and C to A → mutation	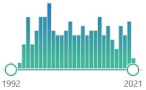 199	0

## Data Availability

Publicly available datasets were analyzed in this study. This data can be found in PubMed^®^ database under search queries described in the manuscript.

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
