# Peer review of "Biomarkers and Mechanisms of Oxidative Stress—Last 20 Years of Research with an Emphasis on Kidney Damage and Renal Transplantation"

_ijms, 2021, doi:10.3390/ijms22158010_

Round 1

Reviewer 1 Report

The references are there, but they are limited to a few paragraphs in page 7 not enough to sustain the title A suggestion would be to reduce the emphasis in the title
The paper is very good extemely documented and useful

Page 3 line 115 figure "areobic '

Page 15, line 535,  P
Maybe in fig. 3 the legend could be arranged alphabetically, thus increasing its readability.
Even if the paper is called" Oxidative Stress and Kidney Damage" the emphasis is on the oxidative stress and a metaanalysis of the publications in the field.
However, the specific kidney points of interest are less addressed, to the point of being limited to a section in page 7. Maybe a shift in the interest in the title would be welcome, as it is a little misleading.

However,the paper is very good and well written. Congratulations!
.

Author Response

Dear Editor-in-Chief, Dear Reviewer

As the corresponding author and on behalf of all the authors, I would like to apply corrections and to answer the issues pointed by reviewers in the manuscript titled “Biomarkers and Mechanisms of Oxidative Stress - last 20 years of research with an emphasis on kidney damage and renal transplantation” (original submission manuscript ID: ijms-1306732) for re-consideration for publication in IJMS MDPI.

Firstly, we would like to thank the reviewer for their valuable comments. We considered all of them, which resulted in the manuscript corrections and extensions. We addressed the reviewers’ (R) questions (Q) with the answers (A) in the order in order of their appearance.

R1Q01: The references are there, but they are limited to a few paragraphs in page 7 not enough to sustain the title. A suggestion would be to reduce the emphasis in the title. Even if the paper is called" Oxidative Stress and Kidney Damage" the emphasis is on the oxidative stress and a metaanalysis of the publications in the field. However, the specific kidney points of interest are less addressed, to the point of being limited to a section in page 7. Maybe a shift in the interest in the title would be welcome, as it is a little misleading.

R1A01: We agree that the subsection concerning “Kidney Damage” is relatively small in a proportion of the entire article. The idea to write a review about oxidative stress was inspired mainly by the essential points of interest studied in our transplantation department. Usually, in our research, in general, we addressed the subject of ischemia-reperfusion injury in kidney transplantation. Sometimes it was limited to the ischemia and kidney storage or the reperfusion and oxidative stress. In others, kidney function was also evaluated in the pre-transplant period; however, the outcome was usually directly or indirectly connected with kidney damage. Thus kidney damage and oxidative stress became a common denominator of searched and discussed issues such as IRI, CRD/ESRD, and KTx. Initially, we started our reference research in kidney transplantation, then it was extended with kidney pathology in general and finally we found it interesting to present the background of oxidative stress. Therefore, subsection titles were given based on the reviewed studies and repeating points of interest. We tried to be consistent and provide complete information about them. Thus subsections grew accordingly, and we agree that metaanalysis of oxidative stress dominated the subject of the kidney. However, we would like to accentuate to a certain extent the emphasis on the kidney in a title, not only due to subsection “3. Oxidative Stress in Kidney Transplantation”, but also due to transplantation reference of all mentioned ROS and biomarkers, which was evaluated by the prism of kidney damage. Based on the above, we decided to propose a more detailed title: “Biomarkers and Mechanisms of Oxidative Stress - last 20 years of research with an emphasis on kidney damage and renal transplantation”. We are aware that much essential work on oxidative stress was done earlier than 20 years ago; however, a lot of those research was a basis for later analyzed studies and some of them were cited and included in a reference.

R1Q02: Page 3 line 115 figure "areobic ' Page 15, line 535,  P

R1A02: “aerobic” was corrected in Figure 1. “Protein” was correted on page 15, line 535.

R1Q03: Maybe in fig. 3 the legend could be arranged alphabetically, thus increasing its readability.

R1A03: Legend was arranged alphabetically.

With best regards

Katarzyna Kotfis MD, PhD, DESA

Reviewer 2 Report

Dear authors

the proposed review describes the relationship between oxidative balance and CKD. The review is interesting, well written and structured.  Few are the researchers who study this complex relationship and because the title focuses on the last 20 years of research I believe there are important works not mentioned:

  • Reference 12: concerning hemodyalisis, there is a recent paper that analyzes the correlation between oxidative stress and inflammation in these patients and CVD risk
  • Concerning 1 section there are many recent studies in animal models (SHR, obesity) in which plasmatic oxidative imbalance and  disfunction in endogenous antioxidants enzyme (GSTP1) is associated with  kidney apoptosis
  • 3 section please correct “Rotein oxidation” with protein oxidation
  • 5 section: regarding regolatory pathway there is an important pathway GSTP1 dimeric form-JNK and apoptosis also decribed in many papers

Author Response

Dear Editor-in-Chief, Dear Reviewer

As the corresponding author and on behalf of all the authors, I would like to apply corrections and to answer the issues pointed by reviewers in the manuscript titled “Biomarkers and Mechanisms of Oxidative Stress - last 20 years of research with an emphasis on kidney damage and renal transplantation” (original submission manuscript ID: ijms-1306732) for re-consideration for publication in IJMS MDPI.

Firstly, we would like to thank the reviewer for their valuable comments. We considered all of them, which resulted in the manuscript corrections and extensions. We addressed the reviewers’ (R) questions (Q) with the answers (A) in the order in order of their appearance.

R2Q01: Few are the researchers who study this complex relationship and because the title focuses on the last 20 years of research I believe there are important works not mentioned: Reference 12: concerning hemodyalisis, there is a recent paper that analyzes the correlation between oxidative stress and inflammation in these patients and CVD risk Concerning 1 section there are many recent studies in animal models (SHR, obesity) in which plasmatic oxidative imbalance and disfunction in endogenous antioxidants enzyme (GSTP1) is associated with kidney apoptosis.

R2A01: Yes indeed, there are newer articles concerning hemodialysis, oxidative stress, and inflammation, including “The DREAM cohort”. They were added to the reference.  In section 1, obesity was not supported with reference. It was added with the emphasis on the role of GSTP1, PI3K, Foxo3a, and other signaling. The connection between apoptosis and intracellular signaling, including GSTP1, was upgraded with additional sentences and citations in section 1.

R2Q02: 3 section please correct “Rotein oxidation” with protein oxidation

R2A02: It was corrected.

R2Q01: df5 section: regarding regulatory pathway there is an important pathway GSTP1 dimeric form-JNK and apoptosis also described in many papers

R2A01: The information about GSTP1 and JNK was added in subsection 5 with proper citations. The Figure 3 was upgraded with appropriate labels and connecting lines. The legend was extended.

With best regards

Katarzyna Kotfis MD, PhD, DESA

Round 2

Reviewer 2 Report

Dear Authors

Thanks for your correction, however I invite you to consider these papers concerning hemodialysis /oxidative stress and inflammation:

PMID:30886674

and these paper regarding Kidney antioxidant disequilibrium and apoptosis injury:

PMID: 28495910

Author Response

Dear Reviewer,

The following information has been added and reference [13] as suggested.

"GSTP1 expression and apoptotic signaling through activation of c-Jun N-terminal kinase (JNK) seem to be mechanisms linking oxidative stress and hypertension in spontaneously hypertensive rats [13]."

The following sentence and reference [17] has been added as suggested.   "Data suggest that cardiovascular complications in patients undergoing hemodialysis are aggravated by oxidative imbalance, which may be a potential target for therapy [17]."

With best regards

Katarzyna Kotfis